# Hand Motion-Aware Surgical Tool Localization and Classification from an Egocentric Camera

**DOI:** 10.3390/jimaging7020015

**Published:** 2021-01-25

**Authors:** Tomohiro Shimizu, Ryo Hachiuma, Hiroki Kajita, Yoshifumi Takatsume, Hideo Saito

**Affiliations:** 1Faculty of Science and Technology, Keio University, Yokohama, Kanagawa 223-8852, Japan; ryo-hachiuma@keio.jp (R.H.); hs@keio.jp (H.S.); 2Keio University School of Medicine, Shinjuku-ku 160-8582, Tokyo, Japan; jmrbx767@keio.jp (H.K.); tsume@keio.jp (Y.T.)

**Keywords:** object detection, surgical tools, open surgery, egocentric camera

## Abstract

Detecting surgical tools is an essential task for the analysis and evaluation of surgical videos. However, in open surgery such as plastic surgery, it is difficult to detect them because there are surgical tools with similar shapes, such as scissors and needle holders. Unlike endoscopic surgery, the tips of the tools are often hidden in the operating field and are not captured clearly due to low camera resolution, whereas the movements of the tools and hands can be captured. As a result that the different uses of each tool require different hand movements, it is possible to use hand movement data to classify the two types of tools. We combined three modules for localization, selection, and classification, for the detection of the two tools. In the localization module, we employed the Faster R-CNN to detect surgical tools and target hands, and in the classification module, we extracted hand movement information by combining ResNet-18 and LSTM to classify two tools. We created a dataset in which seven different types of open surgery were recorded, and we provided the annotation of surgical tool detection. Our experiments show that our approach successfully detected the two different tools and outperformed the two baseline methods.

## 1. Introduction

Recording plastic surgeries in operating rooms with cameras has been indispensable for a variety of purposes, such as education, sharing surgery technologies and techniques, performing case studies of diseases, and evaluating medical treatments [1,2]. Due to the development of mobile hardware, egocentric cameras such as GoPro [3] or Tobii [4] have been introduced to many fields [5] to analyze activities from a first-person perspective while not disturbing the action of the recorder [6].

From the recorded video of the surgery, many tasks have been proposed to analyze surgery, such as workflow analysis [7], phase recognition [8], video segmentation [9], skill assessment [10], and video summarization [11]. The presence and positions of surgical tools are essential information for analyzing surgical procedures [7,8,10,12,13].

Predicting the objects’ rough positions, sizes, and classifications is known as the object detection task. In the computer vision field, the task of object detection is well studied [14,15]. However, these approaches have only been tested with common object detection datasets, such as MS-COCO [16] or Pascal VOC [17], which include the images captured in the daily-life for detecting variety of objects. As a result that there is a huge domain gap between images from everyday life and those associated with plastic surgery, the detection models which are trained on these datasets [16,17] have difficulty detecting the detailed surgical tools with the category information. In the medical vision field, object detection in endoscopic surgery is well studied. Endoscopic surgery is the surgery where the surgeon sees the images through the endoscope camera. An example image of the endoscopic surgery is shown in Figure 1 (left). It can be seen that the surgical tools are clearly seen in the image.

However, in the case of open surgery video captured with the egocentric camera, there are several difficulties in detecting the surgical tools. The example image of open surgery recorded with the egocentric camera is shown in Figure 1 (right). First, the surgical tools are severely occluded by the surgeon’s hand or the other surgical tools. Second, even though there is no occlusion of the tools, it is difficult to classify the tools because their shapes and textures are similar. For example, the overall shapes of the scissors and needle holders (Figure 2) are very similar. Only the tip of them and the grasping part are different, but these parts are mostly occluded by hands or the surgical field.

This paper tackles the detection and classification task of surgical tools from egocentric images for open surgery analysis. A naive approach for detecting the objects is applying a recently proposed deep learning based model, such as Faster R-CNN [15], SSD [19] or YOLO [14]. These methods simultaneously detect and classify the objects from RGB images using their shape and texture information. However, as the textures of the tools are metallic and shiny and shapes are similar among the surgical tools, it might be easy to detect but difficult to classify.

Even though the shapes of the surgical tools are similar, the tools are used for completely different purposes. For example of the scissors and the needle holders, the function of the scissors is to cut the object, but the function of the needle holders is to suture by holding suture materials. However, it is difficult to predict the function of the tools from the images so we assume that the hand motions of the surgeons are different when using different surgical tools. We focus on the motion of the surgeon’s hand to classify surgical tools instead of the textures and shapes of the tools. Figure 3 shows an example of the hand motions associated with the scissors (a) and the needle holders (b). It can be seen that the motion is completely different among tools according to their usage.

We present a framework for detecting and classifying similarly shaped surgical tools, such as scissors and needle holders, from an egocentric video. In the first stage, we detect surgical tools and hands in the image using Faster R-CNN [15]. In this stage, the surgical tools are detected, but are classified as tools but not classified by tool type. In the second stage, as multiple hands such as those belonging to assistants or that do not play a role in the procedure are seen in the image, we select the surgeon’s hand using the maximum overlap ratio between the tools and hands. At the last stage, the sequentially selected hand regions are classified into different tool categories using convolutional neural network (CNN) and long short-term memory (LSTM) module [20].

As there is no dataset available to the public containing open surgery videos via egocentric camera, we recorded our own dataset. The actual plastic surgeries were recorded at our university’s school of medicine. We recorded seven different types of surgery with Tobii cameras [4] attached to the surgeon’s head. We validate our proposed model with this dataset, and we quantitatively evaluate our approach against two baseline methods to verify the effectiveness of our approach. The experiments show that our approach can detect the two surgical instruments separately.

In summary, our contributions are as follows: (1) To the best of our knowledge, this is the first approach to improve the accuracy of surgical tool detection (localization and classification) with similar shape in open surgical videos. (2) We employed hand motion information for classifying surgical tools, instead of tool shape. (3) We created a dataset of a variety of open surgeries recorded with egocentric cameras, we manually annotate detection labels of surgical tools, scissors and needle holders. We conducted extensive experimentation from qualitative and quantitative perspectives to verify the effectiveness of the proposed method.

## 2. Related Work

### 2.1. Object Detection Using Hand Appearance

Many studies have been conducted to improve the accuracy of object detection using information other than that of the target object. For example, Cue et al. [21] developed object detection by estimating the location of target objects from hand segmentation results. They detected hands, and cropped the area where the target object was estimated to be located by using detected hands area. From the resulting cropped image, they classified the target object by using CNN. That is, their method consisted of two modules: the part that estimates the location of the target object and the part that classifies the target object. Ren et al. [22] used optical flow to separate the background from the hand-held objects to improve the accuracy of detection. Since the target videos were first person videos, the calculation of optical flow was greatly influenced by the motion of the body and head. Therefore, they normalized the optical flow by using background motion in the video.

Cue and Ren used hand information to estimate the location of the object, but not to classify or label it. Our study differs from those studies because we used hand motion information to classify tools rather than estimate the position of the target object. In addition, although motion information can be obtained by using optical flow, for first-person video it is necessary to normalize the optical flow as in Ren et al. [22]. However, in our study, hand motion is used in the classification part of object detection, so it is not appropriate to normalize or otherwise alter the raw data. Therefore, the motion information is not obtained by using optical flow; instead, CNN and LSTM [20] are used to extract features from images.

### 2.2. Surgery Video Analysis and Surgical Tool Detection

Studies of surgical tool detection have been mainly focused on procedures in which the surgeon sees the surgical field not directly but through a camera, such as cataract and endoscopic surgery. There are some datasets for the surgical tool detection, such as Cholec80 [23], ITEC [24], M2CAI2016 [18], CATARACT2017 [25] dataset. These datasets cannot be applied to our task as the domain of the surgery in these datasets is different from the domain of the open surgery.

Using these datasets, many methods have been proposed for detecting the surgical tools of the cataract and endoscopic surgery [26,27,28,29] and generating the realistic surgery images [30]. For example, Colleoni et al. [26] proposed a method for detecting the surgical tools and estimating the articulation of the tools for the laparoscopic surgery analysis. Du et al. [27] presented a fully-convolutional network-based articulated 2D pose estimation of the surgical tools for microsurgery video analysis.

In addition, although there are studies about surgical tool detection using the edge and texture information of surgical tools [31,32], many studies have used features obtained by fine-tuning existing CNN algorithms that have been trained with the Imagenet dataset [33]. For example, Twinada et al. [23] fine-tuned Alexnet [34], Roychowdhury et al. [35] fine-tuned Inception-v4 [36], ResNet-50 [37], and NASNet [38], and Raju et al. [39] fine-tuned GoogleNet [40], VGG-16 [41], and Inception-v3. In addition, in the study by Cadene et al. [42], the detection accuracy was improved by inputting the 15-frame presence probability of each tool, obtained by fine-tuning ResNet-200 and Inception-v3, into a hidden Markov process. Thus, there are many studies that take into account temporal information in surgical tool detection. The reason for this is that information about which tools have been used in previous frames helps in identifying which surgical tools are in the current frame. Mishra et al. [43] used LSTM [20] for making it possible to classify tools with multiple labels. In addition, Ai et al. [25] studied surgical tool detection while simultaneously training the CNN and recurrent neural network (RNN) for cataract surgery. In this case, the problem was that as they were simultaneously trained, the loss of the RNN across each frame was not propagated back to the CNN. Therefore, by using the boosting algorithm, they made it possible to train the CNN while and after training the RNN.

In summary, these studies focused on surgeries in which surgeons observed the surgical field through a camera, such as cataract surgery or endoscopic surgery. The relevant work is the work presented by Zhang et al. [44]. They applied RetinaNet [45], which is trained with annotated Youtube open surgery videos, and Simple Online and Realtime Tracking (SORT) algorithm to detect and track the hand during the open surgery. They did not aim to localize and classify the surgical tools.

As this is the first study which aims to detect the surgical tools for open surgery video analysis, we localize and classify the tools which have the similar shapes, scissors, and forceps as seen in Figure 2. As the shapes and textures are similar between two surgical tools, it is difficult to classify these tools using the conventional object detection methods such as Faster R-CNN [15]. In our study, hand appearance features are extracted with ResNet-18 for each frame. Hand motion information is then extracted with LSTM for surgical tool detection labeling.

## 3. Proposed Framework for Surgical Tools Localization and Classification

Our task was to detect the surgical tools that have similar shapes and textures, such as needle holders and scissors. The overview of our surgical tool detection and classification framework is visualized in Figure 4, and the abstract of it is visualized in Figure 5. Our proposed framework consists of three phases: surgical tool and hand localization, selection of the target hand, and surgical tool classification from the motion of the target hand. First, we employ Faster R-CNN to jointly detect the hands and the surgical tools. Note that surgical tools are detected as tools and the specific tool category is not labeled at this stage. Second, as multiple hands can be seen in the image that are irrelevant to the surgical tool classification, we use the overlap ratio between the bounding boxes of the hand operating the tool and the tool itself, respectively, to select the appropriate single hand. Finally, the surgical tool is classified from the sequentially selected target hand frames. We employ ResNet-18 [37] to extract the visual context features for each frame. These features are inputted to LSTM to aggregate the sequential features from which the surgical tool label is predicted. The following explains the details of the proposed framework.

### 3.1. Surgical Tool and Hand Localization

First, we apply Faster R-CNN [15] to detect the surgical tools and hands in the image. Faster R-CNN is one of the major two-staged object detectors. It predicts the object proposal regions using a region proposal network from the image feature encoded with CNN. Then, the refined bounding box and its category are predicted for each object proposal region. Note that all of the surgical tools are detected as tools and each tool category is not predicted in this stage.

### 3.2. Target Hand Selection

Second, we select the single hand which operates the surgical tool in the image. During open surgery, multiple hands from multiple surgeons can be seen in the egocentric image. To classify the category of the surgical tool, the hand which operates the target tool should be selected among multiple detected hand bounding boxes. From the *i*-th hand multiple bounding box $b_{i}^{hand}$, we calculate the overlap ratio $r_{ik}$ with the *k*-th target surgical tool bounding box $b_{k}^{tool}$ as follows:(1)$$r_{ik}\left(b_{i}^{hand},b_{k}^{tool}\right)=\frac{b_{i}^{hand}\cap b_{k}^{tool}}{b_{i}^{hand}\cup b_{k}^{tool}}.$$

To select the single hand bounding box $i^{*}$ which operates surgical tool *k*, the hand bounding box with the maximum ratio is selected:(2)$$i^{*}=\underset{i\in V}{\arg\;\max}\left\{r_{ik}\left(b_{i}^{hand},b_{k}^{tool}\right)\right\},$$
where V denotes the set of bounding boxes labeled as the hand. We use the bounding box $i^{*}$ to classify the surgical tool *k*.

### 3.3. Surgical Tool Classification from Hand Motion

To obtain the motion of the hand, we crop hand image $I_{t}\left(0\leq t\leq T\right)$ which is selected at the previous step. *T* denotes the number of frames for inputting into the LSTM for surgical tool classification. That is, only when the target hand is detected for T consecutive frames, it is used as input data for classification.

#### 3.3.1. Network Architecture

First, from the hand-cropped images $I_{t}$, a visual feature is extracted from each cropped image. In this paper, ResNet-18 [37] is employed as a visual feature extractor, and the cropped image $I_{t}$ is resized to 252×252 (pixels) to make the image size of ResNet-18 input consistent. We extract the visual feature $\psi_{t}\in R^{128}$ from each image $I_{t}$.

After extraction, the visual features of the hands are used as input for LSTM to obtain hand motion information. Then, we aggregate the context features over time, ${\hat{\psi }}_{1}\ldots {\hat{\psi }}_{t}\ldots {\hat{\psi }}_{T}=B\left(\psi_{1},\ldots \psi_{t}\ldots \psi_{T}\right)$, where B is a sequential feature aggregation module that computes the sequential feature ${\hat{\psi }}_{t}$. In our experiments, we employ an LSTM recurrent neural network with one hidden layer for B.

The output feature ${\hat{\psi }}_{1},\ldots ,{\hat{\psi }}_{T}$ is then fed to a multilayer perceptron (MLP) with one hidden layer and rectified linear units (ReLU) activation function [46] to predict the label probability $p_{1},\ldots ,p_{T}$. The sigmoid activation function is applied to the output layer.

#### 3.3.2. Loss Function

We formulate the task of surgical tool detection as the surgical tool and hand detection, and surgical tool classification using the cropped hand images. At this stage, we employ weighted cross-entropy loss for classifying the sequential hand images into the surgical tools for considering the imbalance of images for surgical tool category in the training dataset.

## 4. Experiment

### 4.1. Dataset

As there is no dataset available that contains open surgery recordings, we use Tobii cameras to create our dataset. The surgeries are recorded at Keio University School of Medicine. Video recording of the patients is approved by the Keio University School of Medicine Ethics Committee, and written informed consent is obtained from all patients or their parents. We record seven different types of surgery with Tobii cameras attached to the surgeon’s head. Among the surgical videos, three glove colors, including green, skin-colored, and white, were used, and three surgeons performed the surgery. Five videos were for training and two were for testing. Patient diagnoses included Stahl’s ear, open fracture, lipoma, skin tumor, nevus cell nevus, cryptotia, and cleft palate. Images from each surgery video are shown in Figure 6. Each surgery video is about 20 min long, and was recorded at 25 frames per second (FPS). The frame size of each video is 1920×1080 (pixels).

### 4.2. Faster R-CNN Training

In Phase 1, it is necessary to train the system to detect surgical tools and hands using Faster R-CNN. Therefore, we randomly sampled 2000 images from five surgical videos for training and annotated them with two labels: hand and tool, which did not distinguish between needle holders and scissors, to create a dataset. Using this dataset, we performed tool and hand localization by training in Faster R-CNN. We employ VGG-16 [41] as the backbone of Faster R-CNN, and VGG-16 was pre-trained by using the Imagenet dataset.

### 4.3. Network Training

For Phase 2, we employ an Adam optimizer [47] with a learning rate of $1.0\times 10^{-3}$. When training the model, we randomly sample a data fragment of T=50 frames. The model converged after 15 epochs, which takes about 48 hours on a GeForce RTX 2080. We apply dropout with probability 0.5 during training. We used Keras library (https://github.com/fchollet/keras) to implement the models. The weights of ResNet-18 are initialized with the pretrained ImageNet dataset [33]. The number of frames for inputting into the LSTM was set to 50 because the movie is at 25 fps and 50 frames is equivalent to 2 s. Basic procedures, such as making incisions or suturing, are recognized by humans within 2 s, so 50 frames was appropriate. The number of dimensions of the output of LSTM is 128, and an activation function we employed is the hyperbolic tangent function. We train the model with batch size 5.

### 4.4. Baseline

As a result that there is no other study that has attempted detection of similarly shaped tools in open surgical videos, we defined two baselines by ourselves and performed a comparison experiment.

The first is Faster R-CNN. That is, we compare a method for surgical tool detection that distinguishes two tools with only Faster R-CNN. In the experiment, the same dataset as the proposed method was annotated with scissors and needle holders as separate tools and trained from five surgical videos, and two surgical videos that were not used for training were used for comparison. The detection result is based on whether or not the intersection over union (IOU), with the threshold set to 0.5, is above the threshold. In comparison to this baseline, we validate the effectiveness of using motion information for tool classification.

Second, we distinguish two tools from the single hand-cropped image only. That is, in Phase 3, we verify whether we can distinguish between needle holders and scissors using visual features of the hands without time series information. ResNet-18 was fine-tuned and compared with those pre-trained by the ImageNet dataset. In the experiments, the same data set as the proposed method was annotated with scissors and needle holders as separate tools and trained from five surgical videos, and two surgical videos that were not used for training were used for comparison. Compared to this baseline, we verify that not hand appearance but hand motion (but not appearance) is effective for classification.

### 4.5. Procedure Lengths

In this experiment, we verify the effectiveness of the proposed method by changing the number of frames *T*. That is, we verify how many frames are appropriate as input data. The comparison is made with 25 frames, 50 frames, 100 frames.

## 5. Results and Discussion

### 5.1. Localization and Selection Results

As mentioned earlier, our method is divided into three modules, localization, selection, and classification. Therefore, the lower the accuracy of localization and selection, the lower the accuracy of the proposed method. Using 300 test images from two videos, the test results are shown in Table 1 and Table 2. The correctness of the localization result is measured by whether the IOU, whose threshold is set to 0.5, is above the threshold or not, and the results of the target hand are whether the target hand is selected from detected hands. As shown in Table 1, there are a few cases in which the tools are miss-classified, but the probability of miss-classification over 50 consecutive frames is considered to be very low, which indicates that the recognition of tools (scissors or needle holders) is accurate. As shown in Table 2, the selection of target hand areas is also found to be accurate. As a result that our localization and selection methods have been demonstrated to be accurate, we expect a similar level of accuracy from our proposed classification method.

### 5.2. Proposed Method

The proposed method and the baseline method were trained on five videos and tested on two videos that were not used for training. The number of sequences of scissors were 756, and that of needle holders were 2894. The results are shown in Table 3 and Figure 7. ROC curve and learning curve of the proposed method are shown in Figure 8 and Figure 9.

As show in Table 3, the proposed method is more accurate in detection than the baseline method. Our task is a binary classification, so the chance rate is 0.5. Therefore, the result of only hand means that it is difficult to detect two tools by only hand appearance. Faster R-CNN alone can also make detection with greater accuracy than chance rate, but the value is lower than the proposed method. That is, the proposed method effectively detects scissors and needle holders.

### 5.3. Procedure Length

In this experiment, we verify the effectiveness of the proposed method by changing the number of frames *T*. The comparison is made with 25 frames, 50 frames, and 100 frames. The results are shown in Table 4.

As shown in Table 4, the result of 50 frames has the highest accuracy. In the case of 25 frames, the accuracy is lower than that of 50 frames. The reason for this may be that the surgeon sometimes does not finish suturing the thread with the forceps in one second, and the hand rotation, which is a characteristic movement, may not occur. On the other hand, in the case of 100 frames, the accuracy is close to a chance rate. The reason for this may be that there were too much input data and the learning process was not successful. Moreover, it is also possible that hand movements other than holding the scissors, such as rotational movements, were used by the surgeon for cutting multiple parts of the body with the scissors.

## 6. Conclusions

In this paper, we proposed a method for object detection of similarly shaped tools using hand motion information. The proposed method is divided into three modules, localization, selection, and classification. Faster R-CNN is used for localization and ResNet-18 and LSTM are used for classification. In the experiment, the two tools were detected separately in open surgery videos with an accuracy of 89.5%. In this paper, we detect only two tools by hand motion information. Therefore we would like to develop our method to detect other tools, such as tweezers and forceps, in the future work.

## Figures and Tables

**Figure 1 jimaging-07-00015-f001:**
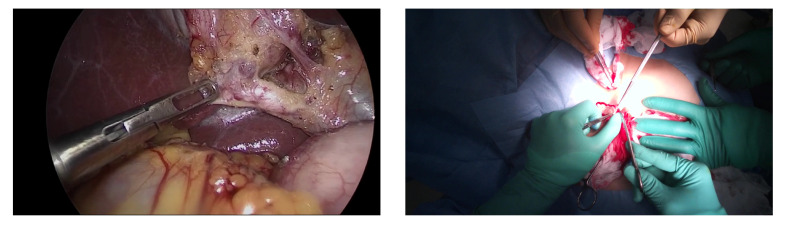
The comparison figure between the images of endoscopic surgery [18] **(left**) and open surgery (**right**).

**Figure 2 jimaging-07-00015-f002:**
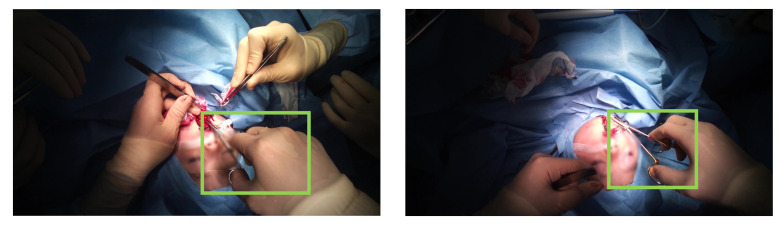
The comparison figure between the scissors (**left**) and the needle holders (**right**).

**Figure 3 jimaging-07-00015-f003:**
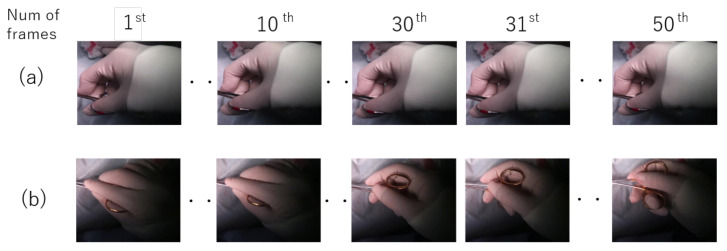
The comparison of the hands’ motion between the surgical tools: (**a**) scissors, (**b**) needle holder.

**Figure 4 jimaging-07-00015-f004:**
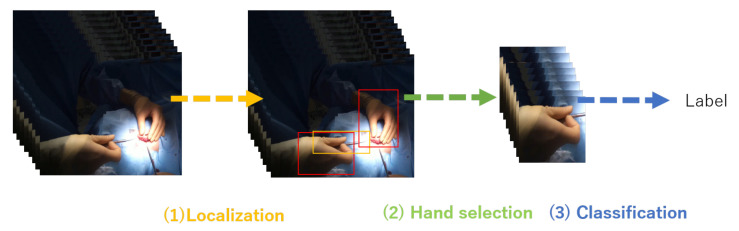
The overview of the proposed surgical tool detection and classification.

**Figure 5 jimaging-07-00015-f005:**
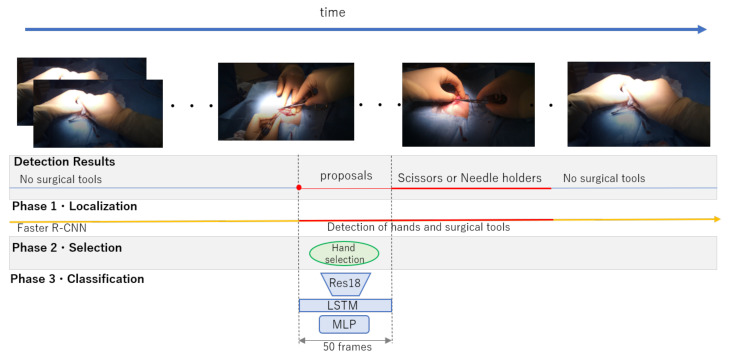
The abstract of the proposed method.

**Figure 6 jimaging-07-00015-f006:**
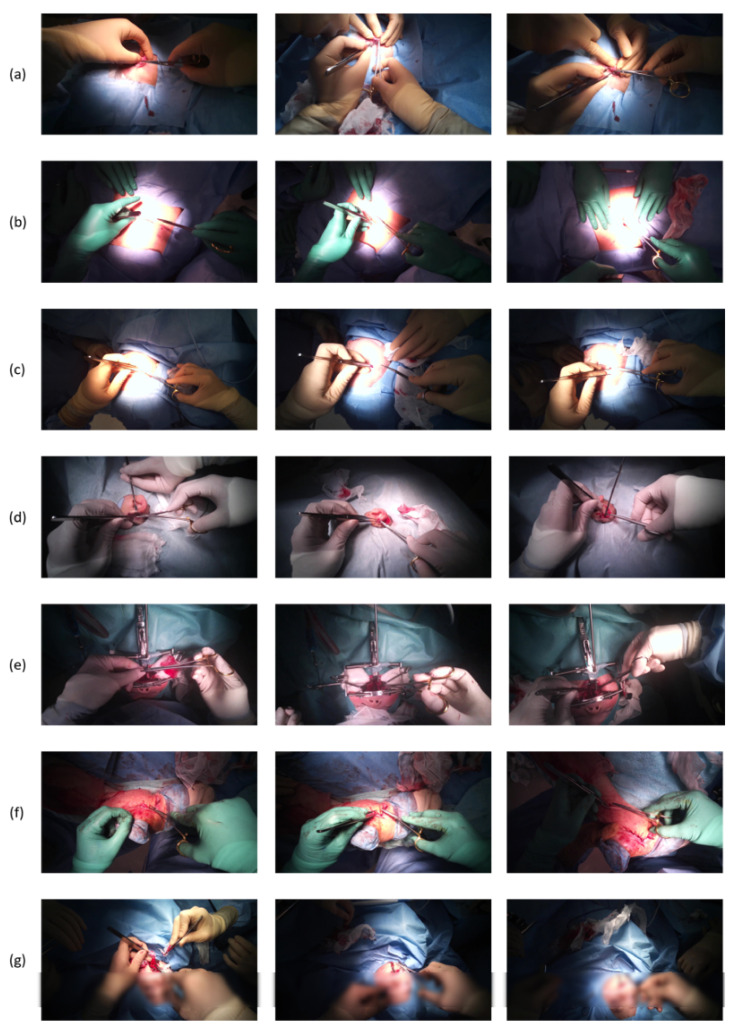
Images of videos we used for the experiment: (**a**) Stahl’s ear, (**b**) lipoma, (**c**) skin tumor, (**d**) cryptotia, (**e**) cleft palate, (**f**) open fracture, and (**g**) nevus cell nevus.

**Figure 7 jimaging-07-00015-f007:**
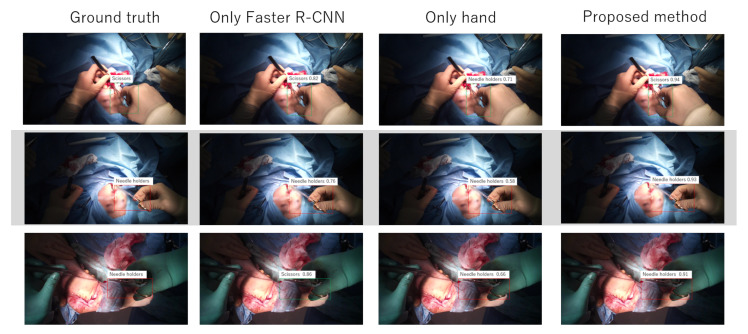
Examples of results.

**Figure 8 jimaging-07-00015-f008:**
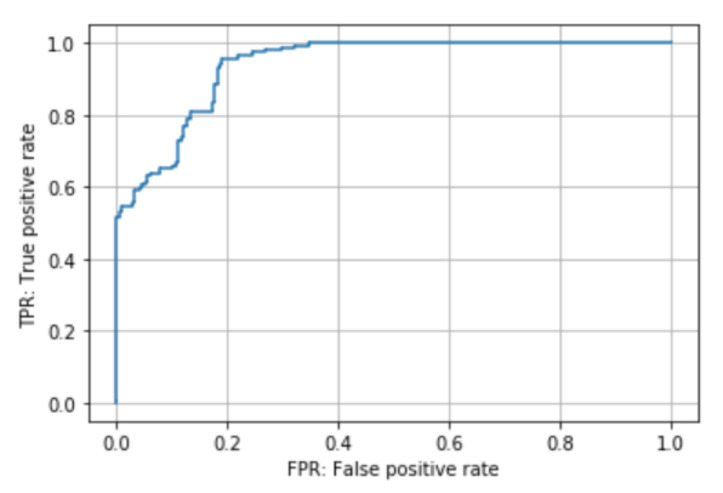
ROC curve.

**Figure 9 jimaging-07-00015-f009:**
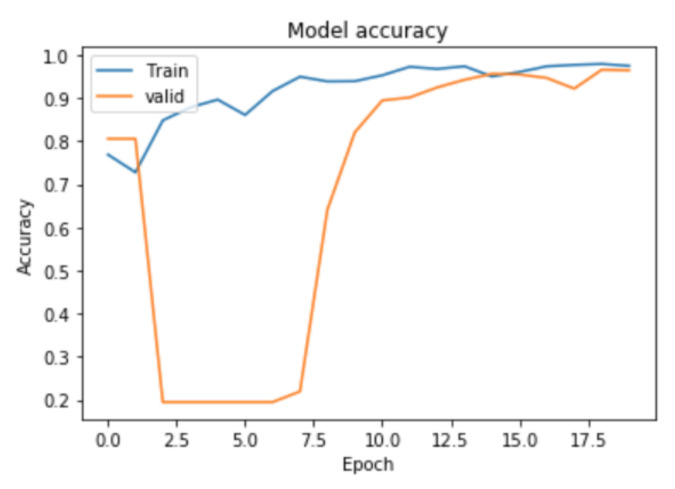
Learning curve.

**Table 1 jimaging-07-00015-t001:** Results of localization.

	AP
tools	0.925
hands	0.713

**Table 2 jimaging-07-00015-t002:** Results of selection.

Method	Accuracy	Recall	Precision	F-Measure
target hand	0.943	0.996	0.946	0.971

**Table 3 jimaging-07-00015-t003:** Results of the proposed method.

Method	Accuracy	Recall	Precision	F-Measure
proposed method	0.895	0.810	0.981	0.888
only Faster R-CNN	0.663	0.811	0.970	0.883
only hand	0.524	0.559	0.230	0.326

**Table 4 jimaging-07-00015-t004:** Results using varying frame numbers in the input data.

Num of Frames	Accuracy	Recall	Precision	F-Measure
25	0.728	0.956	0.703	0.810
50	0.895	0.810	0.981	0.888
100	0.563	0.903	0.500	0.644

## Data Availability

The data presented in this study are available on request from the corresponding author. The data are not publicly available due to privacy protection.

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
