# Peer review of "Hand Motion-Aware Surgical Tool Localization and Classification from an Egocentric Camera"

_2313-433X, 2021, doi:10.3390/jimaging7020015_

Round 1
Reviewer 1 Report
Very interesting topic, and the article was an enjoyable read for me. I believe the article is well-poised to introduce a novel approach in this context.
The manuscript is generally well-written, the objectives and methodology are clear. I have no major concerns overall.
Please consider the following points in the next version.
(1)
The authors make a claim that “there is no dataset available that contains open surgery recordings”. I am not sure if this is entirely accurate, whereas I could find similar datasets that are publicly available as well.
Examples include:
ITEC SurgicalActions160 Dataset
http://ftp.itec.aau.at/datasets/SurgicalActions160/
MVOR Dataset, and Cholec80 Dataset
http://camma.u-strasbg.fr/datasets
But perhaps those datasets differ from what the authors were seeking for, I hope the authors would elaborate on that point in their response.
(2)
Another claim made by the study is that “there is no other study that has attempted detection of similarly shaped tools in open surgical”.
Well, in general I recommend using a more tentative language while making claims about the literature. Also, I am wondering about the work below, and I hope the authors would be able to describe how these applications are different.
Zhang, M., Cheng, X., Copeland, D., Desai, A., Guan, M. Y., Brat, G. A., & Yeung, S. (2020). Using Computer Vision to Automate Hand Detection and Tracking of Surgeon Movements in Videos of Open Surgery. arXiv preprint arXiv:2012.06948.
Sarikaya, D., Corso, J. J., & Guru, K. A. (2017). Detection and localization of robotic tools in robot-assisted surgery videos using deep neural networks for region proposal and detection. IEEE transactions on medical imaging, 36(7), 1542-1549.
Wang, S., Xu, Z., Yan, C., & Huang, J. (2019). Graph convolutional nets for tool presence detection in surgical videos. In International Conference on Information Processing in Medical Imaging (pp. 467-478). Springer, Cham.
(3)
The authors mentioned the dataset development as part of their contributions.
However, do they actually intend to publish that dataset?
(4)
The authors touched briefly on the imbalance in the training dataset.
Please could you specify how many samples were included for each category in the dataset?
That said, have you considered applying some augmentation techniques to mitigate that imbalance?
(5)
For clarity of results, please include these figures:
ROC curve
Learning curve plotting the loss in train and test sets.
(6)
Please could you mention which library was used to implement the models (e.g. TensorFlow, PyTorch)?
It would be appreciated if the authors could share the implementation of their models.
(7)
The manuscript currently lacks a ‘Discussion’ section to elaborate further on the results, and how this connects to the literature. This would also allow a good space for discussing possible limitations and directions for future work.
(8)
Please revise the Alexnet reference in line 103.
Reviewer 2 Report
The manuscript "Hand Motion-Aware Surgical Tool Localization and Classification from an Egocentric Camera" presents a deep-learning method for surgical tool detection in RGB images acquired during open surgery. The manuscript addresses a relevant problem but presents several weaknesses that have to be solved.
Introduction
The authors should clarify how the surgeon's hands are detected already at line 61.
The camera is attached tot the surgeon hand. Is this comfortable for the surgeons? Does this setup alter the natural movement of the hand? A survey on the system usability with at least 3 surgeons should be performed.
Why were only 2 surgical instruments considered?
State of the art
The authors should cite work on surgical tool analysis, for example:
- Colleoni, Emanuele, et al. "Deep learning based robotic tool detection and articulation estimation with spatio-temporal layers." IEEE Robotics and Automation Letters 4.3 (2019): 2714-2721.
- Du, Xiaofei, et al. "Articulated multi-instrument 2-D pose estimation using fully convolutional networks." IEEE transactions on medical imaging 37.5 (2018): 1276-1287.
- Marzullo, Aldo, et al. "Towards realistic laparoscopic image generation using image-domain translation." Computer Methods and Programs in Biomedicine (2020): 105834.
Sec. 3.1
Did you train Faster R-CNN from scratch? Clarify this here.
What do you mean by "we crop T frames cropped hand image"?
The authors have to describe the LSTM architecture otherwise it is impossibile to replicate the work.
Sec. 4
How many different surgeons were included in the study?
What do the authors mean by different surgery?+
Which was the frame size?
Did you perform any frame selection?
Were the instruments always present in the camera field of view?
In 4.2, which was the optimizer? The loss function? How many epochs were set?
In 4.3, did this data fragment of 50 frames always show the instruments?
Line 188: add a reference relevant to the procedure length (2 seconds).
In 4.4, it is not clear if you use the same data split as in the previous experiments.
Sec. 4.4 should be strongly improved. Now it is not clear 1) how were the experiments performed, 2) which was the motivation behind.
Section 5
In Sec. 5.1, how many times both instruments were present in the 300 images? How many frames did not show any instrument?
As a general comment, it is not clear why the testing dataset is different for the different steps of the proposed method. The authors should always consider the same training/ validation/ testing dataset.
The discussion of the results is completely missing.
Round 2
Reviewer 2 Report
Thank you for addressing my comments.